# Elucidation of the Characteristics of Soil Sickness Syndrome in Japanese Pear and Construction of Countermeasures Using the Rhizosphere Soil Assay Method

**Tomoaki Toya [1],\*, Masayoshi Oshida [2], Tatsuya Minezaki [3], Akifumi Sugiyama [4], Kwame Sarpong Appiah [5]** 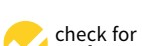**, Takashi Motobayashi [5],\*** **and Yoshiharu Fujii [5]**

1    Chiba Prefectural Chiba Agricultural Office, 473-2 Ookanazawa-cho, Midori-ku, Chiba 266-0014, Japan
2    Chiba Prefectural Agriculture and Forestry Research Center, 180-1 Ookanazawa-cho, Midori-ku, Chiba 266-0014, Japan; m.oshd@pref.chiba.lg.jp
3    Ajinomoto Healthy Supply Co., Inc., 1-15-1 Kyobashi, Chuo-ku, Tokyo 104-0031, Japan; Tatsuya_minezaki@ajinomoto.com
4    Research Institute for Sustainable Humanosphere (RISH), Kyoto University, Gokeshou, Uji-city, Kyoto 611-0001, Japan; akifumi_sugiyama@rish.kypto-u.ac.jp
5    Department of International and Innovative Agriculture Science, Tokyo University of Agriculture and Technology, 3-5-8 Saiwai-cho, Fuchu, Tokyo 183-8538, Japan; ksappiah90@gmail.com (K.S.A.); yfujii@cc.tuat.ac.jp (Y.F.)
*    Correspondence: t.ty5@pref.chiba.lg.jp (T.T.); takarice@cc.tuat.ac.jp (T.M.); Tel.: +81-080-43-300-0950 (T.T.)

**Abstract:** The continuous planting of Japanese pear leads to a soil sickness syndrome that eventually affects the growth and yield of the plant. In this study, we aimed to elucidate the characteristics of soil sickness syndrome in the Japanese pear and construct countermeasures using the rhizosphere soil assay method that can quantify the risk of soil sickness syndrome by inhibitory chemicals. Water flushing treatment, rainfall treatment, and the incorporation of test soils with different rates of activated carbon were evaluated on the risks of soil sickness. The water flushing treatment under laboratory conditions and exposure of the continuous cropping soil to rainfall in the open field decreased the inhibition rate of the soil. The decrease in soil inhibition rate was presumed to be the result of accumulated growth inhibitory substances in the soil being washed away by water. In addition, activated carbon with the potential to reduce the soil sickness syndrome was selected using the rhizosphere soil assay method. It was clarified that the mixing of the selected activated carbon with the continuous cropping soil reduced the inhibition rate and increased the growth of pear trees increased compared to the untreated soil from the continuous cropping field. The inhibition rate of the soil from the continuous cropping field was reduced to the level of soil with no history of Japanese pear cultivation. In the replanted field, these treatments can promote the growth of trees by reducing the influence of soil sickness syndrome.

**Keywords:** activated carbon; adsorption; allelopathy; growth inhibitory substances; inhibition rate of soil; treatment to flush water; tree growth

## 1. Introduction

The Japanese pear (*Pyrus pyrifolia* (Burm. F.) Nakai) is a representative fruit tree in Japan and is cultivated mainly in Chiba prefecture near the capital, Tokyo [1]. However, after 30–40 years of continuous planting, the growth of Japanese pear trees has declined, and the yield has almost halved. About 50 years have passed since the main varieties of Japanese pear became widespread and it has become necessary to replant new trees in many producer fields. In the replanted fields of Japanese pears, the initial growth has halved due to soil sickness syndrome which is an obstacle to the promotion of replanting [2]. Soil sickness syndrome is caused by auto-toxic compounds accumulated in the soil through root exudation by the previous crop. The compounds then reduce the growth of crops planted

later by 20% to 50% [3]. Soil sickness syndrome has been observed in many fruit trees, and it has been reported that hydrocyanic acid compounds are involved in peaches [4] and Japanese apricots [5]. In the Japanese pear, the growth inhibitory substances that accumulate in the soil to cause inhibition have not been clarified and no countermeasures have been taken or proposed.

The rhizosphere soil assay is a method for evaluating the risk of occurrence of soil sickness syndrome in replanted fields using lettuce as a receptor plant [6,7]. Toya et al. [8] reported that the rhizosphere soil assay method can be used to quantify the risk of soil sickness in the continuous cropping soil of the Japanese pear (the pear soil). In the Japanese pear, growth inhibition began to be seen when the soil inhibition rate was 30% or more, growth was suppressed by about 40%, and growth was halved at 60% [8]. In addition, by utilizing this method, it was confirmed that the mixing of Japanese pear roots into the soil did not cause soil sickness syndrome and that the accumulation of the growth inhibitor substances in soil occurred during the growth of the pear trees [9]. In this way, it was speculated that there is a high possibility that the characteristics of the Japanese pear soil sickness syndrome can be elucidated by using the rhizosphere soil assay method. Therefore, in this study, we sought to clarify using the rhizosphere soil assay method whether water and rainfall treatments or the mixing of activated carbon with pear soil could reduce the risk of soil sickness.

The auto-toxic compounds released from alfalfa have been reported to be water-soluble substances that migrate through soil [10]. Additionally, the growth inhibitory substances of the Japanese pear are presumed to be water-soluble [11] and flushing them out of the soil by water treatment may reduce the risk of soil sickness. Therefore, in Experiment 1, we examined whether the pear soil treated with water would reduce the risk of soil sickness under laboratory conditions. Experiment 2 examined whether the risk of soil sickness of pear soil could be reduced by exposing the soil to rainfall in an open field.

Activated carbons, due to their large capacity to adsorb biochemical compounds, have been used effectively to reduce the chemical interference of allelochemicals [12,13]. The growth of tomatoes in hydroponics was inhibited by organic substances that are exudated from roots but were eliminated by adding activated carbon [14]. In Japan, activated carbon was used to adsorb the growth inhibitory substances in asparagus [15]. However, there are various types of activated carbon, and the effects vary depending on the type of crop [16,17]. To that effect, it is necessary to select the most suitable material to reduce the soil sickness syndrome of Japanese pear soil. Therefore, in Experiment 3, several types of activated carbons were mixed with the pear soil and evaluated by the rhizosphere soil assay method to select activated carbons suitable for the Japanese pear. In Experiment 4, the treatment volume of the selected activated carbon and the elapsed time required after the treatment were examined. Furthermore, in Experiment 5, the effect on tree growth was investigated by mixing activated carbon with the pear soil and cultivating Japanese pear saplings.

As a result of summarization, we elucidated some of the characteristics of growth inhibitory substances in the Japanese pear and based on these findings, we constructed measures to mitigate soil sickness syndrome.

## 2. Materials and Methods

### 2.1. Test Soil

The Japanese pear was cultivated at the Chiba Prefectural Agriculture and Forestry Research Centre. Pots (volume 22.5 L) were filled with soil (Volcanic ash soil) collected from the vegetable field and Japanese pear saplings of the cultivar "Akizuki" were planted for five months. Fertilizer was applied by dividing 100 g/tree of chemical fertilizer (Nitrogen: Phosphorous: Potassium (NPK) 15:15:15) into each early part of April to July. Watering was done daily with a nozzle installed so that the soil did not dry out. After the trees were pulled, the remaining soil was used as the test material (the pear soil). As the non-pear soil, the soil collected from the vegetable field without the cultivation of pear sapling was used.

## 2.2. Rhizosphere Soil Assay Method

The risk of soil sickness was evaluated using the rhizosphere soil assay method [6]. Test soils were dried at 60 °C for 24 h with a drying machine (MOV-112F, Sanyo Electric Biomedica Co. Ltd., Tokyo, Japan), crushed and passed through a 2 mm sieve. Three grams of the soil was placed in each well of the culture multi-dish (6 holes, NUNC). Five mL of low-temperature gelled agar (0.75%, (Nacalai Tesque Inc., Kyoto, Japan) autoclaved at 115 °C for 15 min) was added, mixed and hardened, and then 5 mL of agar was layered. A well filled with only agar was prepared and used as a blank control. Lettuce seeds ("Legacy", Takii Seed Co. Ltd., Kyoto, Japan) were sown on the agar and kept at 25 °C for 3 days under dark conditions. After that, the root length of the lettuce was measured. The measurement was performed 5 times in each section. The soil inhibition rate was calculated as a percentage of the blank control as shown in the formula below.

$$z = (x - y)/x \times 100 \tag{1}$$

where z: soil inhibition rate (%), x: average value of blank root length, y: average value of root length of test soil.

## 2.3. Effect of Water Treatment on Growth Inhibitory Activity of Pear Soil (Experiment 1)

The pear soil or non-pear soil (100 g each) were put in a plastic case (length:width:depth = 170 mm:120 mm:70 mm). The plastic case had five holes with a diameter of 1 mm so that water can drain. An amount of 100 mL of pure water was poured over the soil once, thrice or five times every two days from 4 December 2019. The inhibitory activity of the pear soil and the non-pear soil (untreated control) was evaluated by using the rhizosphere soil assay method. Each section was repeated 3 times.

## 2.4. Effect of Rainfall on the Inhibition Rate of the Pear Soil (Experiment 2)

In the pear soil plot, 6 L of the pear soil was filled in a pot (volume 12.2 L, diameter 30 cm). As a control, a section filled with only non-pear soil was set up. The pots were left in the open field for six months, and each plot had three replicates. No saplings were planted in the pots and no management other than weeding was performed. The soil in the pot was agitated and 100 g of each was collected (0, 2, 4, and 6 months after set-up) and the inhibition rate was measured by the rhizosphere soil assay method. The precipitation data during this period was taken from Japan Meteorological Agency (AMEDAS Sakura).

## 2.5. Evaluation of the Effectiveness of Different Activated Carbons on Reducing Soil Sickness Syndrome (Experiment 3)

The test soils (200 g each) were placed in a plastic case and mixed with activated carbons. The activated carbons used in this study were activated carbon A (Granular dojo-saiseitan, with tree as material, pellet, Ajinomoto Healthy Supply Co., Inc., Tokyo, Japan), activated carbon B (Yashikol, with coconut husk as material, granular, Ohira Chemical Industry Co., Ltd., Osaka, Japan), activated carbon C (Brocol C, coal-based activated carbon, Ohira Chemical Industry Co., Ltd., Osaka, Japan), activated carbon D (powder, with tree as material, Aminol Chemical Research Institute, Hyogo, Japan), and activated carbon E (Shirasagi MW50, with tree as material, fine granules, Osaka Gas, Osaka, Japan). Two grams (1% by weight) of each of the activated carbons were mixed thoroughly with the soil in the plastic cases. As a control, pear soil without activated carbon and a non-pear soil plot was established.

The plastic cases were covered with a polyethene bag to maintain soil moisture and kept at 25 °C under dark conditions. Six weeks after mixing the soil with activated carbon, the soil was collected after stirring and the soil inhibition rate was evaluated by the rhizosphere soil assay method.

*2.6. Evaluation of the Effectiveness of Selected Activated Carbon on Soil Sickness Syndrome (Experiment 4)*

Activated carbon A (Ajinomoto Healthy Supply Co., Ltd.) had a high effect of reducing the soil inhibition rate in Experiment 3 and was further evaluated. The test soils (200 g each) were placed in plastic cases and mixed with different rates of activated carbon. Amounts of 2 g, 4 g, and 6 g of the activated carbon (1, 2, and 3% by weight) were mixed with the pear soil. As controls, the pear soil without activated carbon and non-pear soil was established. The experimental setup was repeated 5 times in each plot. The plastic cases were covered with a polyethylene bag to maintain soil moisture and kept at 25 °C under dark conditions. The soil in the pot was agitated, 50 g each was collected on 0, 3, and 6 weeks after mixing, and the inhibitory effects of the soils were evaluated by the rhizosphere assay method.

*2.7. Effect of Activated Carbon Treatment on the Growth of Japanese Pear Trees (Experiment 5)*

The cultivation test was conducted at Chiba Prefectural Agriculture and Forestry Research Center. An amount of 200 g of activated carbon (based on Experiment 4) was mixed with 20 kg of the pear soil. After filling the pot (volume 22.5 L), water was added, and the pot was allowed to stand indoors for 1 month (activated carbon treatment). The other treatments were the pear soil and non-pear soil without activated carbon. Each of the treatments had 5 replications. Japanese pear sapling cultivars "Akizuki" were then planted. Fertilization and watering were the same as described in Section 2.1.

At the time of planting, 50 g of soil was collected from the pots of each plot and the soil inhibition rate was evaluated in the same manner in Experiment 1. The main trunk diameter of the tree at planting was measured 10 cm above the grafted part. The tree growth survey and dismantling survey 5 months after planting. The shoot was surveyed for lengths of 10 cm or more. The length, the diameter at 5 cm above the base, and the number of occurrences were measured. The total elongation was the sum of the lengths of all the new shoots. The number of leaves and leaf color were measured. For leaf color, 20 leaves of each tree were randomly selected and measured with a chlorophyll meter (SPAD-502, Konica Minolta Co., Ltd., Tokyo, Japan). The dry weight of each organ of the tree was evaluated separately for a new shoot, stem, leaf and root (below the graft). All samples were measured after drying at 90 °C for 1 week. In the activated carbon plot, one tree died due to White root rot and data were excluded for that plot.

## 3. Results

*3.1. Effect of Water Treatment on the Inhibition Rate of Japanese Pear Soil*

The effect of water treatment on the inhibition rate of pear soil on lettuce was evaluated using the rhizosphere soil assay method and the results are shown in Table 1. When the frequency of water treatment was 1 and 3 every two days, the soil inhibition rates were 65.6% and 57.5%, respectively, and were not significantly different from that of the control (64.2%). When the frequency of water flushing was increased to 5 times every two days, the inhibition rate (52.1%) was significantly lower than that of the control and the one-time flush treatment.

**Table 1.** Effect of water treatment on the inhibition rate of the pear soil.

| The Number of Processing Times | Soil Inhibition Rate(%) | |
| :---: | :---: | :---: |
| | **Pear Soil** | **Non-Pear Soil** |
| 0 (Control group) | 64.2 a | 16.8 |
| 1 | 65.6 a | 21.6 |
| 3 | 57.5 ab | 15.5 |
| 5 | 52.1 b | 18.6 |
| *p*-value | <0.01 | 0.68 |

Data were analyzed by Tukey–Kramer method after arcsine transformation. Different letters indicate a significant difference at the 5% level.

### 3.2. Effect of Rainfall on the Inhibition Rate of Japanese Pear Soil

The effect of rainfall on the inhibition rate of Japanese pear soil was evaluated using the rhizosphere soil assay method. The soil inhibition rate (67.6%) at two months after set-up was not significantly different from the inhibition rate of 64.4% at the beginning of the experiment (Figure 1). However, four months after the set-up, the inhibition rate was significantly reduced to 43%. The pear soil inhibition rate further decreased to 37% by the sixth month although not significantly different from the inhibitory effect observed after four months. The cumulative precipitation was 380 mm two months after the set-up, 960 mm after four months, and 1170 mm after six months. The volume of water in the pot after 2, 4, and 6 months were 45 L, 114 L, and 139 L, respectively, when converted to 10 L of soil.

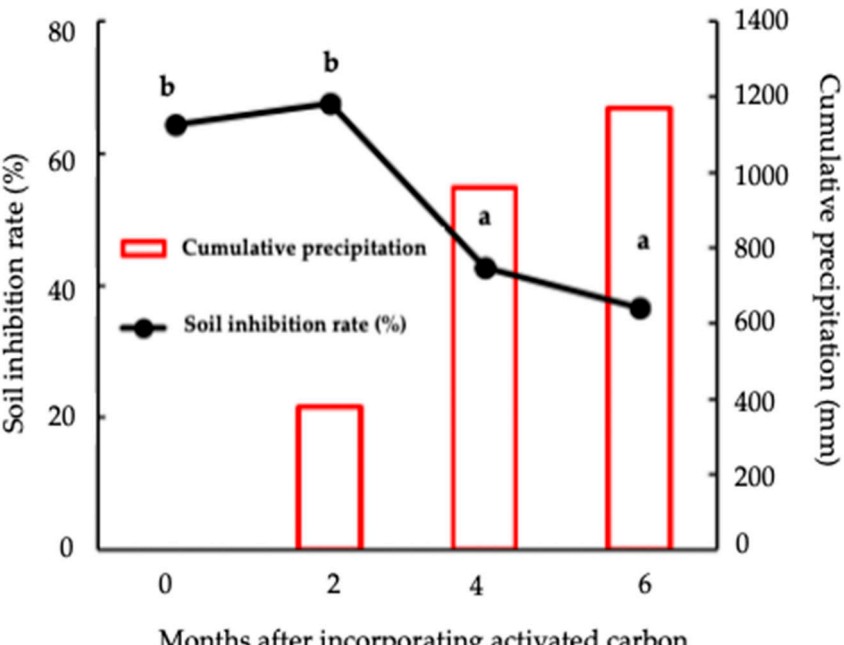

**Figure 1.** Changes in soil inhibition rate in the pear soil left in the open field and cumulative precipitation. Data were analyzed by Tukey–Kramer method after arcsine transformation. Different letters indicate a significant difference at the 5% level.

### 3.3. Effectiveness of Different Activated Carbons Reducing Soil Sickness Syndrome

The inhibition rate of soil was 41.4% in the activated carbon A, which was significantly lower than 66.1% in the untreated pear soil plot, but higher than 24.6% in the non-pear soil plot (Table 2). The other activated carbon treatments had a soil inhibition rate of 60.4% to 67.8%, which was not significantly different from the untreated pear soil. From the above, activated carbon A was more effective at reducing the risk of soil sickness and was further evaluated at different rates of application. There was a strong correlation ($r = 0.93$; significant at 1% level) between the inhibition rate of pear-soil and cumulative rainfall over the six-month period.

**Table 2.** Selection of activated carbon suitable for Japanese pear using the rhizosphere soil assay method.

| Treatments | Soil Inhibition Rate (%) |
|---|---|
| Activated carbon A | 41.4 b |
| Activated carbon B | 60.4 a |
| Activated carbon C | 66.6 a |
| Activated carbon D | 63.9 a |
| Activated carbon E | 67.8 a |
| Pear-soil | 66.1 a |
| Non-pear soil | 24.6 c |
| *p*-value | <0.01 |

Data were analyzed by Tukey–Kramer method after arcsine transformation. Different letters indicate a significant difference at the 5% level.

### 3.4. Effect of the Selected Activated Carbon Treatment on Soil Sickness Syndrome

It was clarified that the mixing of activated carbon reduces the inhibition rate of soil at different incorporation rates over time. After 1 day of mixing, the soil inhibition rate was in the range of 59.8% to 65.4% at an activated carbon application rate of 1–3% treatments, which was the same as 65.9% in the untreated pear soil plot (Figure 2). Three weeks after the mixing, the inhibition rate of the activated carbon incorporated at 2% and 3% were 54.4% and 55.7%, respectively, which were significantly lower than the pear soil plot of 62.8%. Six weeks after the mixing, the inhibition rate of the activated carbon incorporated at 1, 2, and 3% were 49.9%, 41.2% and 35.2% respectively, which were significantly lower than 68.0% in the untreated pear soil.

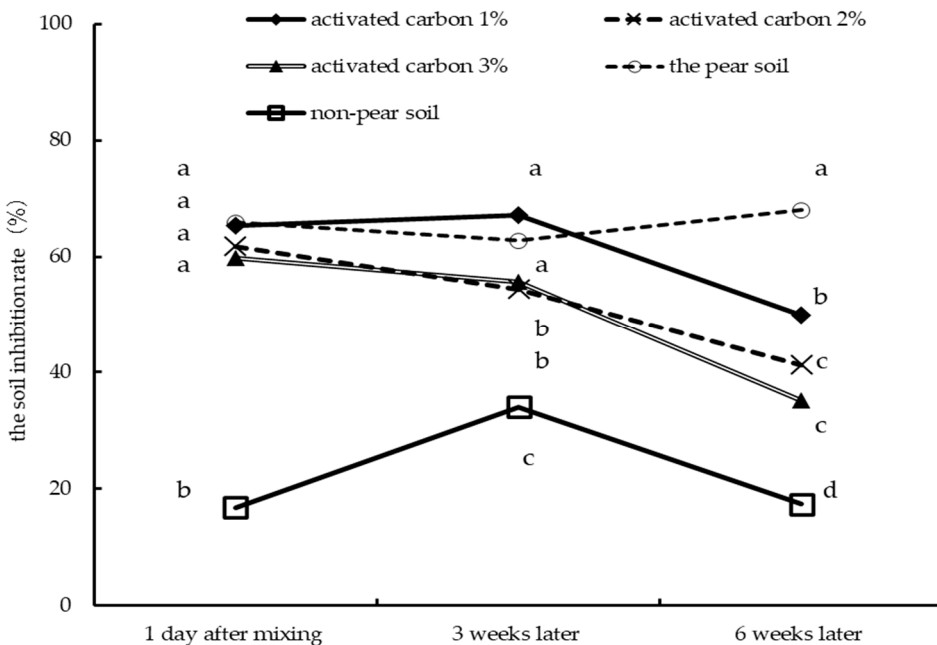

**Figure 2.** Transition of the inhibition rate of soil mixed with activated carbon. Data were analyzed by Tukey–Kramer method after arcsine transformation. Different letters indicate a significant difference at the 5% level.

### 3.5. Effect of Activated Carbon Incorporation on the Growth of Japanese Pear Trees

The soil inhibition rate at planting was 37.2% in the activated carbon treatment, which was significantly lower than 62.8% in the pear soil, but significantly higher than 23.4% in the non-pear (Table 3). There was no difference in the main trunk diameter of the tree at planting.

**Table 3.** Soil inhibition rate at the time of planting and main trunk diameter of Japanese pear saplings.

| Treatments | Soil Inhibition Rate (%) | Main Trunk Diameter (mm) |
|---|---|---|
| Activated carbon | 37.2 b | 15.2 |
| Pear soil | 62.8 c | 15.9 |
| Non-pear soil | 23.4 a | 14.9 |
| *p*-value | <0.01 | 0.12 |

Data were analyzed by Tukey–Kramer method after arcsine transformation. Different letters indicate a significant difference at the 5% level.

The tree growth at the end of cultivation is shown in Table 4. The number of shoots in the activated carbon treatment was 8.5 branch/tree, which was significantly higher than the pear soil plot (6.0 branch/tree), and the same as 8.0 branch/tree in the non-pear soil plot. The total shoot elongation of 6.5 m/tree in the activated carbon plot was significantly higher than the pear soil plot (3.8 m/tree), and about the same as the 5.3 m/tree in the non-pear soil plot. The number of leaves in the activated carbon treated plot (236.8 leaves/tree) was significantly higher than the untreated pear soil plot (156.2 leaves/tree) and the same as in the non-pear soil plot (239.6 leaves/tree). For other survey items, no significant difference was found in the treatment plot.

**Table 4.** The growth of trees in soil mixed with activated carbon.

| Treatments | Shoot | | | | Leaf | | Main Trunk Diameter (mm) |
|---|---|---|---|---|---|---|---|
| | Number (Branch/Tree) | Length (cm) | Total Elongation (m/Tree) | Proximal Diameter (mm) | Number (No/Tree) | SPAD Values | |
| Activated carbon | 8.5 b | 77.1 | 6.5 b | 8.7 | 237 b | 48.7 | 22.2 |
| Pear soil | 6.0 a | 61.5 | 3.8 a | 8.5 | 156 a | 47.8 | 20.5 |
| Non-pear soil | 8.0 b | 67.3 | 5.3 ab | 8.7 | 240 b | 46.8 | 21.6 |
| *p*-value | <0.01 | 0.24 | <0.01 | 0.92 | <0.01 | 0.47 | 0.13 |

Different letters indicate a significant difference at the 5% level by the Tukey–Kramer method.

The dry weights of pear trees are shown in Table 5. The dry weight of shoot in the activated carbon treated plot (168 g/tree) was significantly heavier than the pear soil plot (89 g/tree) and the same as that of the non-pear soil plot (141 g/tree). The dry weight of leaves in the activated carbon plot was 111 g/tree, which was significantly heavier than the pear soil plot (71 g/tree) and the same as that of the non-pear soil plot (85 g/tree). The dry weight of roots in the activated carbon treated plot (245 g/tree) was significantly heavier than the pear soil plot (184 g/tree) and not different from the non-pear soil plot.

**Table 5.** The dry weight of trees in soil mixed with activated carbon.

| Treatment | Shoot | | Stem | Leaf | | Root | | Whole Tree | |
|---|---|---|---|---|---|---|---|---|---|
| | (g/Tree) | | | | | | | | |
| Activated carbon | 168 | b | 202 | 111 | b | 245 | b | 727 | b |
| Pear soil | 89 | a | 177 | 71 | a | 184 | a | 520 | a |
| Non-pear soil | 141 | b | 214 | 85 | ab | 248 | b | 688 | b |
| *p*-value | <0.01 | | 0.95 | <0.01 | | 0.02 | | <0.01 | |

Different letters indicate a significant difference at the 5% level by the Tukey–Kramer method.

The whole tree biomass in the activated carbon treated plot (727 g/tree) was significantly higher than the untreated pear soil plot of 520 g/tree, and the same as that of the non-pear plot

## 4. Discussion

In this study, we used the rhizosphere soil assay to elucidate the characteristics of Japanese pear soil sickness syndrome and to mitigate the replant failure. First, it was clarified that the risk of soil sickness was reduced by flushing water through the pear soil under laboratory conditions. With the assumption that the specific gravity of the soil used in this test was 0.7 [18], the volume required to significantly reduce the soil inhibition rate was about 100 L per soil 10 L. From this, it was assumed that the growth inhibitory substances from Japanese pear are water-soluble and can be washed away by rainfall even in the replanted field.

Therefore, in Experiment 2, it was investigated whether the risk of soil sickness could be reduced by leaving the field open and exposing it to rainfall. The inhibition rate of the pear soil decreased with increasing precipitation. The soil inhibition rate did not decrease when the volume of water calculated from precipitation was 45 L per 10 L of soil, however, the soil inhibition rate decreased significantly when the volume of water reached 114 L. This volume of water was about the same as the result of the laboratory test in Test 1. In addition, it was considered that the reason for the decreased soil inhibition rate as the cumulative precipitation increased was due to the flushing away of the growth inhibitory substances by water as demonstrated in Experiment 1. On the other hand, under natural conditions, the soil inhibition rate was as high as 40% even when exposed to rainfall for 6 months. When Japanese pear saplings were planted in soil with an inhibition rate of 40%, there was a high possibility that growth suppression will occur due to soil sickness syndrome [8]. In actual fields, it has been confirmed that there is a high risk of land up to a depth of 40 cm [8]. It is presumed that it will take more time than the results of this study for water to penetrate to this depth and for the risk of soil sickness to decrease. A study was conducted to evaluate a replanted field using a hot water drip treatment machine [19] that can easily flush a large volume of water. As a result, it was reported that the inhibition rate of soil was reduced, and the growth of replanted trees was improved by the treatment [20]. This method can reduce the control of White root rot, which is also a problem at the time of replanting, as well as reduce soil sickness syndrome.

In this study, the activated carbon manufactured by Ajinomoto Healthy Supply Co., Ltd. was highly effective in reducing the risk of soil sickness of Japanese pear soil. Activated carbon with a growth-promoting effect has been confirmed in Japanese apricots, and the appropriate incorporation rate was 1% [17]. On the other hand, there was no growth-promoting effect in peach when activated carbon was incorporated in a replanted field [21]. In the Japanese pear, a study was conducted using activated carbon which had effects on asparagus [16] but there was no clear effect [22]. On the other hand, the characteristics of activated carbon differ greatly depending on the material and treatment temperature [23], so it is necessary to select activated carbon suitable for the crop. Subsequently, the selected activated carbon was tested by varying the application rates. As a result, it was clarified that it took 6 weeks for the inhibition rate of the pear soil to be significantly reduced even under favorable temperature and moisture conditions. However, it is likely to take more time than the setting of this study when activated carbon is used in an open field because the replanting of Japanese pears is carried out in autumn and winter when it is dry with low temperatures. In this study, the incorporation of activated carbon at a rate of 3% was excellent in reducing the inhibition rate of the pear soil. On the other hand, increasing the amount of activated carbon leads to an increase in cost. In this study, a cultivation test was conducted with 1% of the standard amount. As a result, the growth of the pear trees in activated carbon treated soil increased compared to the untreated pear soil, and soil sickness syndrome was sufficiently reduced. The amount of activated carbon applied may differ depending on the amount of carbon in the soil and the content of inorganic components [24], and it is necessary to clarify the quantity to be treated. Additionally, in an experiment in which 0.8% of activated carbon was added to soil in which *Rehmannia glutinosa* was continuously cultivated, vanillic acid in the soil decreased

and growth improved [25]. In the Japanese pear, the growth inhibitory substances have not been clarified and the adsorption effect of activated carbon could not be confirmed directly.

The growth inhibitory substances of soil sickness syndrome have been reported in various tree species. In apples, phenolic acid is the cause, and the concentration of phenolic acid in the soil from apple orchards is highest in the soil layer of 0~30 cm in spring [26]. In peach, during the growth cycle, the accumulation of autotoxins released from the root into soil strongly restrains the perennial tree growth in the same soil plot and the major toxic cyanide was only detected in the water extracts of root bark not in the root wood part [27]. In the Japanese pear, the growth inhibitory substances have not been identified, however, analysis by the rhizosphere soil assay method has clarified that it is abundant in the surface layer with many roots [8] and that it is released during growth like that of apple and peach. In addition, from the results of this study, it is clarified that the growth inhibitory substances are water-soluble and can be adsorbed by activated carbon. From these facts, by utilizing the method, the characteristics of the growth inhibitor substances of soil sickness syndrome can be clarified, which can lead to the establishment of countermeasures.

## 5. Conclusions

It was clarified by using the rhizosphere soil assay that, the growth inhibitory substances of the Japanese pear are water-soluble and can be washed away by rainfall even in the replanted field. In addition, the selected activated carbon reduced the inhibition rate of Japanese pear soil and hence can minimize soil sickness syndrome. In the future, we plan to clarify the effectiveness of water treatment and mixing of activated carbon in field tests and to identify the growth inhibitory substances of pear.

**Author Contributions:** Conceptualization, T.T., M.O. and Y.F.; methodology, T.T., T.M. (Takashi Motobayashi) and Y.F.; soft-ware, T.T.; validation, T.T., K.S.A. and A.S.; formal analysis, T.T. and M.O.; investigation, T.T. and M.O.; resources, T.M. (Tatsuya Minezaki) and A.S.; data curation, T.T.; writing—original draft preparation, T.T.; writing—review and editing, K.S.A., Y.F. and T.M. (Takashi Motobayashi); visualization, T.T.; supervision, Y.F. and T.M. (Takashi Motobayashi); project administration, M.O., A.S. and Y.F. All authors have read and agreed to the published version of the manuscript.

**Funding:** This work was partly supported by JST CREST Grant Number JPMJCR17O2 and JSPS KAKENHI Grant Number 26304024.

**Institutional Review Board Statement:** Not applicable.

**Informed Consent Statement:** Not applicable.

**Data Availability Statement:** Not applicable.

**Conflicts of Interest:** The funders had no role in the design of the study; in the collection, analyses, or interpretation of data; in the writing of the manuscript, or in the decision to publish the results.

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
