# Peer review of "Elucidation of the Characteristics of Soil Sickness Syndrome in Japanese Pear and Construction of Countermeasures Using the Rhizosphere Soil Assay Method"

_agronomy, doi:10.3390/agronomy11081468_

Round 1

Reviewer 1 Report

The comments and suggestions on the manuscript entitled" Elucidation of the Characteristics of Soil Sickness Syndrome in Japanese Pear and Construction of Countermeasures Using the Rhizosphere Soil Assay Method has been appended below—I found the article is highly interesting to go through however, it was very hard to follow the methodology section as too many experiments were performed, however, it makes confusion and hard to follow, I hope authors can work on methods to make it more compatible to reader-friendly ---

  1. Abstract- as per the Journal format, there must be a background methodology and results followed by conclusions and future implications in a running paragraph – However, I could not see any relevant result included and future implications were also lagging – please modify as per the format of the journal so that abstract represents the whole zest of the paper---
  2. Methods –Authors are advised to use days instead of writing the date. It does not fit well. Please replace with days post-treatment or pretreatment like that ----
  3. The methodology section is a bit hard to understand and follow; I think it can be organized in a better way so that it will be easy to understand and follow. I strongly suggest please modify the whole methodology section to make it more reader-friendly---

Author Response

a

Reviewer 2 Report

Hi, Authors,

I am glad to read such interesting study and I appreciate the work and time you put into this paper. Please see my comments below:

  • Overall, the paper is well written. I had a few minor suggestions.
  • M&M – 2.3 – what are the differences between those activated carbons? I think the readers will be interested in learning the details of those activated carbons, such as their size? A table would help explain the details.
  • Line 129 – delete ‘was used’
  • Results 3.2 – have you run a correlation analysis to find out whether there is a relationship between cumulative precipitation and soil inhibition rate? I suggest try it.
  • Line 199 – please add ‘different’ between ‘not significantly’ and ‘from’
  • Line 205 – please paraphrase ‘This was the volume of water in the pot, which was 45 L, 114 L, and 139 L, respectively, when converted to 10 L of soil.’ This sentence readiness can be improved.
  • Results 3.3 – delete ‘The effectiveness to reduce soil sickness syndrome in pear soil by different activated carbon treatment was evaluated using the rhizosphere soil assay method.’ This is repetitive as you mentioned it in the M&M.
  • Results 3.3 – it’s important to clearly state that A is more effective than others. The tables showed the results, but the writing didn’t state it clearly.
  • Results 3.4 – delete ‘Activated carbon (Ajinomoto Healthy Supply Co., Ltd) was more effective at reducing the risk of soil sickness and was further evaluated at different rates of application.’ This is repetitive as you mentioned it in the M&M.
  • Results 3.4 – Paraphrase this sentence and put it as the first sentence. ‘From these facts, it was clarified that the mixing of activated carbon reduces the inhibition rate of soil at different incorporation rates over time.’ This is an overall conclusion that should be placed in the beginning.
  • Results 3.5 – delete ‘The effect of incorporating activated carbon in pear soil on the growth of Japanese pear trees was investigated.’ This is repetitive as you mentioned it in the M&M.
  • Discussion
    • Overall, discussion is thoughtful, however, I think more in-depth discussion should be included. This can be done by comparing with more similar studies, unless the study in this field is limited. The authors should read more similar studies and incorporated them into the discussion section.
    • Line 317-344 – the structure of this paragraph can be improved. Below are some minor suggestions, authors should read the discussion several more times and modify as needed
    • Line 324 – delete ‘Therefore, effective activated carbon was selected in a laboratory test using the rhizosphere soil assay method.’ This is repetitive.
    • Line 330 – extra space between ‘conditions and It’.
    • Line 325 – This sentence should be the first sentence for this paragraph. ‘As a result, activated carbon manufactured by Ajinomoto Healthy Supply Co., Ltd. was highly effective in reducing the risk of soil sickness of Japanese pear.’ In my opinion, it’s better to give an overall discussion point first before moving to details.
    • Paraphrase ‘In this study, the incorporation of activated carbon treated was 3%, which was excellent in reducing the inhibition rate of the pear soil.’
  • Conclusion – more suggestions are suggested, particularly for pear growers, such as what and when a grower should do to make sure replanting success can be improved maximally. In addition, what are future directions of such studies? There must be something needed to be explored extending from this study.
  • Graphs and Tables – make sure the format is consistent. For example, ‘NS’ is included in Table 1 but not others. Choose a format and stick to it.

Author Response

Dear Reviewer 2

Please see the file.

Thank you. 
